# Has Incineration Replaced Recycling? Evidence from OECD Countries

**Thomas C. Kinnaman** [1,*] **and Masashi Yamamoto** [2]

1   Department of Economics, Bucknell University, Lewisburg, PA 17837, USA
2   Department of Economics, Tokai University, Tokyo 151-8677, Japan
*   Correspondence: kinnaman@bucknell.edu

**Abstract:** Despite public efforts to promote recycling, recycling rates in OECD countries with mature recycling programs in 2000 have largely stagnated over the past 20 years. Could the rapid growth in incineration have played a role in suppressing the growth in recycling? This paper introduces a model to understand the relationship (possibly positive or negative) between incineration and recycling. A cursory examination of the data within individual countries appears to support the model by demonstrating both a positive and negative relationship. An econometric model that estimates incineration has a negative but statistically insignificant effect on recycling rates. This result contrasts with the previous literature which found a negative and statistically significant negative relationship between incineration and recycling.

**Keywords:** solid waste; recycling; incineration; waste management policy; circular economy

## 1. Introduction

The average recycling rate across the thirty-five membership countries that comprise the Organization for Economic Development (OECD) increased from 19.3% in 1990 to 26.2% in 2018 [1]. This increase could be attributed to various policy measures, such as the European Union's 1999 Landfill Directive and the Circular Economy Package that established a goal of recycling 65% of municipal waste by 2030. However, recycling rates among the OECD's top ten recycling countries in 2000 (Austria, Belgium, Finland, Germany, Netherlands, Norway, South Korea, Switzerland, Sweden, and the United States), and thus the most mature programs today, increased only marginally from 27% in 2000 to 32% in 2018. Few explanations for the stagnation in recycling rates have emerged.

Could the recent rise in solid waste incineration capacity have played a role? The EU's 1999 Landfill Directive imposed binding goals for reducing the use of landfills, and incineration rates subsequently increased. The percentage of total waste incinerated in European OECD countries nearly doubled from 14% in 1996 to 27.6% in 2018. Incineration has grown most rapidly in Norway (where incineration has increased from 13% in 1996 to 51% in 2018), Finland (from 2% to 57%), Estonia (from 0% to 44%), Italy (from 5% to 21%), and Portugal (from less than 1% to 19%).

However, incinerators require large quantities of waste fuel to sustain high combustion temperatures. Most recycled materials, such as paper, cardboard, plastic, and other household items are excellent potential sources of fuel. If these materials are increasingly being incinerated, then the growth of incineration might have unexpectedly hampered efforts to increase recycling quantities in OECD countries, as has been found in Japan [2]. This paper utilizes annual OECD member waste data to estimate the effect of incineration on the percentage of solid waste that is recycled and landfilled. We estimate that an increase in the number of incinerators decreases the percentage of all waste that is landfilled and insignificantly decreases the percentage recycled.

## 2. The Relevant Literature

The question of whether incineration is replacing recycling is an important one if the external costs associated with incineration exceed those of recycling. Incinerator emissions can contribute to climate change, acidification, eutrophication, and threaten human health [3,4]. Incinerators can also generate electricity and various building materials [5]. Boesch et al. [3] estimate a ton of incinerated waste emits up to 382 kg of carbon dioxide emissions. This number can drop to 303 kg if the generated energy replaces energy from coal combustion. Recycling programs also generate external costs via collection and material processing activities and generate external benefits as recycled materials displace the use of raw materials in manufacturing [6]. Accounting for all of these factors, Kinnaman et al. [7] estimate the net social costs associated with incineration are about USD 30 per ton and the social cost of recycling can be as low as USD 500 per ton. Given this large difference, the question of whether incineration may reduce recycling becomes rather important for social efficiency. Regarding incineration and landfilling, Dijkgraaf et al. [8] find that the external costs of incineration are less than those for landfilling. Alternatively, O'Donovan et al. [9] find incineration less socially costly than landfilling.

To our knowledge, only one paper has estimated the direct relationship between incineration and recycling. Using panel data from a sample of large Japanese municipalities, Yamamoto et al. [2] find that municipalities with incinerators with "excess capacity" (a defined binary variable) experience a 2% to 4% reduction in recycling rates relative to similar municipalities without excess incineration capacity. Municipalities with excess incineration capacity may choose to combust recycled materials to maintain constant temperatures, which of course would result in reduced recycling rates. We extend this literature by investigating whether this negative relationship between incineration and recycling extends beyond Japan—a country that features by far the highest rates of incineration in the world.

Although we are aware of only this single paper that estimates the effect of incineration on recycling, we believe this paper may also contribute to the economics literature that estimates household recycling rates. Economic papers have considered both underlying household characteristics and exogenous economic variables when attempting to understand differences in recycling rates across households or jurisdictions. For example, Aadlan and Caplan [10] surveyed households directly and found that the average household is willing to pay USD 5.61 per month to have access to curbside recycling services. Note that these households faced no economic incentives to recycle—instead, recycling appears to be based on underlying household preferences. Koford et al. [11] also estimate households are willing to pay for access to curbside recycling (USD 2.29 per month). Halvorsen [12] associates these recycling preferences to a social recycling norm. Using British data, Abbott et al. [13] find no statistically significant relationship between recycling rates and the costs of recycling programs or any program attributes associated with curbside recycling programs, such as frequency of collection or number of materials accepted. Instead, they also identify a social norm to explain recycling rates.

Thus, even though recycling rates vary across jurisdictions, the only empirical explanation found for this variation appears to be the underlying recycling preferences of households. Economic variables, such as the cost of waste disposal or the price of recycled materials do not appear to affect recycling rates. If this paper finds a statistical link between incineration and recycling rates, then an economic factor—the increased availability of a substitute for recycling (incineration)—would explain recycling rates.

## 3. Modeling Waste Disposal Decisions

In this section, we develop a model for why incineration and recycling might be positively or negatively related. Assume cost-minimizing waste allocation decisions are made by a municipal, a regional, or a state government (for simplicity, we will use the term municipality throughout this paper). Assume the municipality is endowed with technologies to landfill (with quantity $Q_L$), incinerate ($Q_I$), and recycle ($Q_R$) an exogenous quantity ($\widetilde{Q}$) of solid waste. Thus $Q_L + Q_I + Q_R = \widetilde{Q}$. At some point in the past, the

municipality planned for waste disposal by exogenously investing in incineration facilities designed to have a life span of several decades. Let $\overline{Q}$ denote this exogenous incineration capacity. Note that $\overline{Q}$ can be less than or greater than $\widetilde{Q}$, but obviously $\overline{Q} > Q_I$.

Assume the total cost of incineration is $TC_I = W(Q_I)$. Assume incineration costs are also a function of excess incineration capacity $(\overline{Q} - Q_I)$. Excess incineration capacity can be helpful or unhelpful. Helpful excess incineration capacity is desirable to allow for periodic maintenance and to accommodate unexpected peaks in waste quantities. Assume such helpful excess incineration capacity reduces the total cost of incineration by $C(\overline{Q} - Q_I)$. However, owing to the details of incineration technology, too much excess capacity is costly. Unhelpful excess incineration capacity requires furnaces to be shut down while waiting for waste to accrue [5]. The incinerator must temporarily store waste and periodically ignite and extinguish furnaces rather than sustain consistent combustion temperatures. Intermittence also complicates the process of removing pollutants and dioxins from the air stream because temperatures must pass through the 200 to 600 degree Celsius danger zone where dioxin formation is most conducive [4]. Incinerators typically devote effort to minimize the time involved with passing through this thermal zone. Define the costs associated with unhelpful excess incineration capacity as $E(\overline{Q} - Q_I)$. Therefore, the overall cost of incineration is

$$TC_I = W(Q_I) - C(\overline{Q} - Q_I) + E(\overline{Q} - Q_I)$$

where $W' > 0$, $C' > 0$, $E' > 0$, $W'' > 0$, $C'' > 0$, and $E'' > 0$.

Recall that each municipality is also endowed with landfilling and recycling technologies. The total cost of these two options is given by $TC_L = L(Q_L)$ and $TC_R = R(Q_R)$, respectively, where the first ($L'$ and $R'$) and second ($L''$ and $R''$) derivatives are both positive to reflect positive and rising marginal costs.

The goal of the municipality is to choose the quantity incinerated, landfilled, and recycled to minimize the total costs of managing $\widetilde{Q}$ units of waste material, where again $Q_I + Q_L + Q_R = \widetilde{Q}$. Substituting for $Q_L$ gives the following objective,

$$MIN \ W(Q_I) - C(\overline{Q} - Q_I) + E(\overline{Q} - Q_I) + L\left(\widetilde{Q} - Q_I - Q_R\right)$$
$$+ R(Q_R).$$

The first-order conditions (with respect to changes in $Q_I$ and $Q_R$) for this minimization process are

$$W'(Q_I) + C\prime(\overline{Q} - Q_I) - E'(\overline{Q} - Q_I) - L'\left(\widetilde{Q} - Q_I - Q_R\right) = 0$$
$$-L'\left(\widetilde{Q} - Q_I - Q_R\right) + R'(Q_R) = 0.$$

The second-order conditions for cost minimization are $W'' - C'' + E'' + L'' > 0$, $R'' + L'' > 0$, and $(W'' - C'' + E'' + L'')(R'' + L'') - L''L'' > 0$. This latter condition is useful for interpreting the signs of comparative statics generated below

Solutions to the cost-minimization problem can be written as general functions of the exogenous variables in the model, $Q_I = Q_I^*(\overline{Q})$ and $Q_R = Q_R^*(\overline{Q})$. Substituting these solutions back into the first-order conditions and differentiating with respect to $\overline{Q}$ allows us to express the change in incineration, recycling, and (via further substitution) landfilling from an exogenous increase in incineration capacity. These comparative statics are:

$$\frac{\partial Q_I}{\partial \overline{Q}} = \frac{(-C'' + E'')(R'' + L'')}{(W'' - C'' + E'' + L'')(R'' + L'') - L''L''} <> 0 \tag{1a}$$

$$\frac{\partial Q_R}{\partial \overline{Q}} = -\left(\frac{L''}{(R'' + L'')}\right)\frac{\partial Q_I}{\partial \overline{Q}} <> 0 \tag{1b}$$

$$\frac{\partial Q_L}{\partial \overline{Q}} = -\left( \frac{R''}{(R'' + L'')} \right) \frac{\partial Q_I}{\partial \overline{Q}} <> 0 \tag{1c}$$

Note the indifferent signage in each comparative static. We are interested in (1b)—how an exogenous increase in incineration capacity affects recycling. However, first we need to interpret the sign and magnitude of (1a). The denominator of (1a) is of course positive for cost minimizing decisions. The sign of the numerator depends entirely on the sign of $(-C'' + E'')$. These two terms represent the slopes of the marginal costs associated with helpful and unhelpful excess incineration capacity, respectively. If the change in the marginal benefit of helpful excess capacity $(-C'')$ is small relative to the change in the marginal cost of unhelpful excess capacity $(E'')$, then adding incineration capacity will increase the overall marginal cost of incineration above the marginal costs of landfilling and recycling. Total costs are of course minimized when all marginal costs are equal. The marginal cost of incineration can be reduced by decreasing the quantity incinerated. Thus, $\frac{\partial Q_I}{\partial \overline{Q}} < 0$ when $(-C'' + E'') < 0$, and recycling and landfilling increases. This counter intuitive result—that adding incineration capacity decreases incineration—may take time to evolve. Perhaps it is helpful to imagine a municipality with an excessively large incinerator constructed some decades in the past and very expensive to operate due to the lack of waste fuel necessary to maintain consistent combustion temperatures. This municipality might find it more expensive to employ this large incinerator than a similar municipality with a smaller incinerator.

The opposite logic is true when $(-C'' + E'') > 0$. Adding incineration capacity in this case generates helpful excess capacity, which decreases the overall marginal cost of incineration below other waste management methods. Equating these marginal costs is accomplished by increasing the quantity incinerated. Thus, $\frac{\partial Q_I}{\partial \overline{Q}} > 0$ when $(-C'' + E'') > 0$. Note that the magnitude of these two changes depend upon how sensitive the marginal costs of incineration, landfilling, and recycling are to changes in their respective quantities ($W''$, $L''$, and $R''$).

The sign $\frac{\partial Q_R}{\partial \overline{Q}}$ in Equation (1b) will be the opposite of the sign of Equation (1a). Basically, any change in incineration from adding capacity will be accompanied by an opposite change in recycling. $\frac{\partial Q_R}{\partial \overline{Q}} > 0$ if $C'' < E''$ and $\frac{\partial Q_R}{\partial \overline{Q}} < 0$ if $C'' > E''$. The magnitude of these changes depends on how responsive the marginal costs of landfilling and recycling are to their respective quantities (the size of $L''$ and $R''$).

To summarize, if $C'' < E''$, then adding incineration capacity increases incineration costs causing incineration to fall and recycling to rise. However, if $C'' > E''$, then adding excess capacity reduces incineration costs causing incineration quantities to rise and recycling quantities to fall. In this latter case, incinerating recyclable materials averts high costs associated with managing unwanted excess incineration capacity. Thus, the theory provides two possible directions for the causation between incineration and recycling. What do the data say?

## 4. The Data

The OECD provides annual data on the percentage of waste landfilled, recycled, and incinerated each year by each member country [1]. Although some countries have reported such data for decades, data reporting became widespread in most countries beginning in about 1990. Table 1 summarizes the percentage landfilled, recycled, and incinerated for all OECD countries in 1990 (or the first available year) and again in 2010 (the first year utilized in the estimation below), and again in 2018. In 1990, OECD countries on average reported landfilling about 61% of all waste, incinerating about 19%, and recycling about 15%. The omitted category in Table 1 is composted waste. The large water content associated with most compostable waste makes such waste not the ideal fuel source for incineration. Incineration rates include both those facilities that generate energy and those that do not. In 2018, the most recent year when all countries reported data, landfilling decreased to just 42% of all waste, incineration increased slightly to 22%, and recycling

increased to 25%. Notice that most of the recycling change occurred prior to 2010—recycling rates increased at a slower rate after 2010.

**Table 1.** Incineration rates, recycling rates, and landfill rates in OECD countries.

| Country | Incineration (%) | | | Recycling (%) | | | Landfill (%) | | |
|---|---|---|---|---|---|---|---|---|---|
| | 1990 | 2010 | 2018 | 1990 | 2010 | 2018 | 1990 | 2010 | 2018 |
| Australia | 0 | 8 | 9 | 40 | 40 | 42 | 52 | 52 | 49 |
| Austria | 9 | 36 | 39 | 28 | 28 | 26 | 3 | 3 | 2 |
| Belgium | 45 | 41 | 43 | 5 | 35 | 34 | 44 | 2 | 1 |
| Canada | 4 | 4 | 4 | 19 | 18 | 20 | 74 | 72 | 69 |
| Chile | 0 | 0 | 0 | 0 | 0 | 0 | 100 | 100 | 100 |
| Czech Republic | 0 | 16 | 17 | 0 | 14 | 27 | 100 | 68 | 49 |
| Denmark | 63 | 52 | 53 | 9 | 29 | 27 | 20 | 3 | 1 |
| Estonia | 0 | 0 | 44 | 0 | 12 | 26 | 99 | 78 | 20 |
| Finland | 2 | 22 | 59 | 30 | 20 | 27 | 65 | 45 | 1 |
| France | 36 | 35 | 36 | 7 | 20 | 24 | 49 | 29 | 22 |
| Germany | 17 | 33 | 31 | 23 | 46 | 49 | 51 | 0 | 0 |
| Greece | 0 | 0 | 1 | 6 | 15 | 15 | 94 | 83 | 80 |
| Hungary | 7 | 10 | 16 | 0 | 16 | 27 | 93 | 70 | 49 |
| Iceland | 18 | 8 | 4 | 6 | 16 | 26 | 72 | 71 | 57 |
| Ireland | 0 | 4 | 30 | 8 | 35 | 34 | 92 | 57 | 26 |
| Israel | 0 | 0 | 1 | 0 | 9 | 7 | 88 | 89 | 78 |
| Italy | 5 | 18 | 21 | 4 | 20 | 31 | 90 | 49 | 26 |
| Japan | 72 | 76 | 79 | 5 | 19 | 20 | 20 | 1 | 1 |
| Korea | 2 | 22 | 25 | 5 | 60 | 59 | 94 | 18 | 15 |
| Latvia | 0 | 0 | 3 | 0 | 9 | 19 | 100 | 91 | 31 |
| Lithuania | 0 | 0 | 19 | 0 | 4 | 24 | 100 | 94 | 33 |
| Luxembourg | 53 | 36 | 45 | 13 | 27 | 28 | 27 | 18 | 7 |
| Mexico | 1 | 0 | 0 | 0 | 4 | 5 | 1 | 96 | 95 |
| Netherlands | 15 | 49 | 44 | 35 | 25 | 26 | 8 | 2 | 1 |
| Norway | 14 | 50 | 53 | 8 | 27 | 29 | 78 | 6 | 4 |
| Poland | 0 | 0 | 24 | 0 | 18 | 27 | 98 | 74 | 42 |
| Portugal | 0 | 19 | 21 | 0 | 11 | 12 | 87 | 62 | 50 |
| Slovak Republic | 9 | 11 | 10 | 2 | 6 | 21 | 70 | 78 | 61 |
| Slovenia | 0 | 1 | 14 | 2 | 21 | 53 | 97 | 60 | 13 |
| Spain | 9 | 9 | 13 | 15 | 18 | 18 | 52 | 62 | 54 |
| Sweden | 41 | 51 | 53 | 13 | 34 | 31 | 44 | 1 | 0 |
| Switzerland | 49 | 50 | 48 | 22 | 34 | 31 | 23 | 0 | 0 |
| Turkey | 0 | 0 | 0 | 0 | 0 | 9 | 93 | 99 | 90 |
| United Kingdom | 9 | 13 | 37 | 7 | 25 | 27 | 83 | 46 | 17 |
| United States | 15 | 12 | 13 | 14 | 26 | 26 | 69 | 54 | 53 |
| OECD—Europe | 15 | 23 | 27 | 11 | 24 | 28 | 67 | 40 | 29 |
| OECD—Total | 19 | 19 | 22 | 15 | 23 | 25 | 61 | 45 | 42 |

These global averages mask wide variations in landfilling, incineration, and recycling across individual OECD countries within each year. For example, among just European OECD countries, incineration increased from 15% to 27% over the entire time interval while incineration decreased in the United States and a few other countries. Several countries in Northern Europe exceeded incineration rates of 50% in 2018. Some OECD countries recycled small amounts in 2018 (Mexico at 5%), several countries incinerated less than 5% in 2018, and landfill rates varied substantially among countries in the EU where Germany and Finland landfilled less than 1% and Greece landfilled 80%.

Figure 1 plots the percentage incinerated and the percentage landfilled for every OECD country in every year from 1990 to 2018. These two forms of waste management appear negatively correlated. If the econometric modelling below estimates a causal relationship—that increases in incineration and reduces in landfilling—then we might conclude that the increase in incineration has helped European countries at least partially

satisfy the requirements of the EU's Landfill Directive (even in the presence of possible relaxation options for non-complying countries).

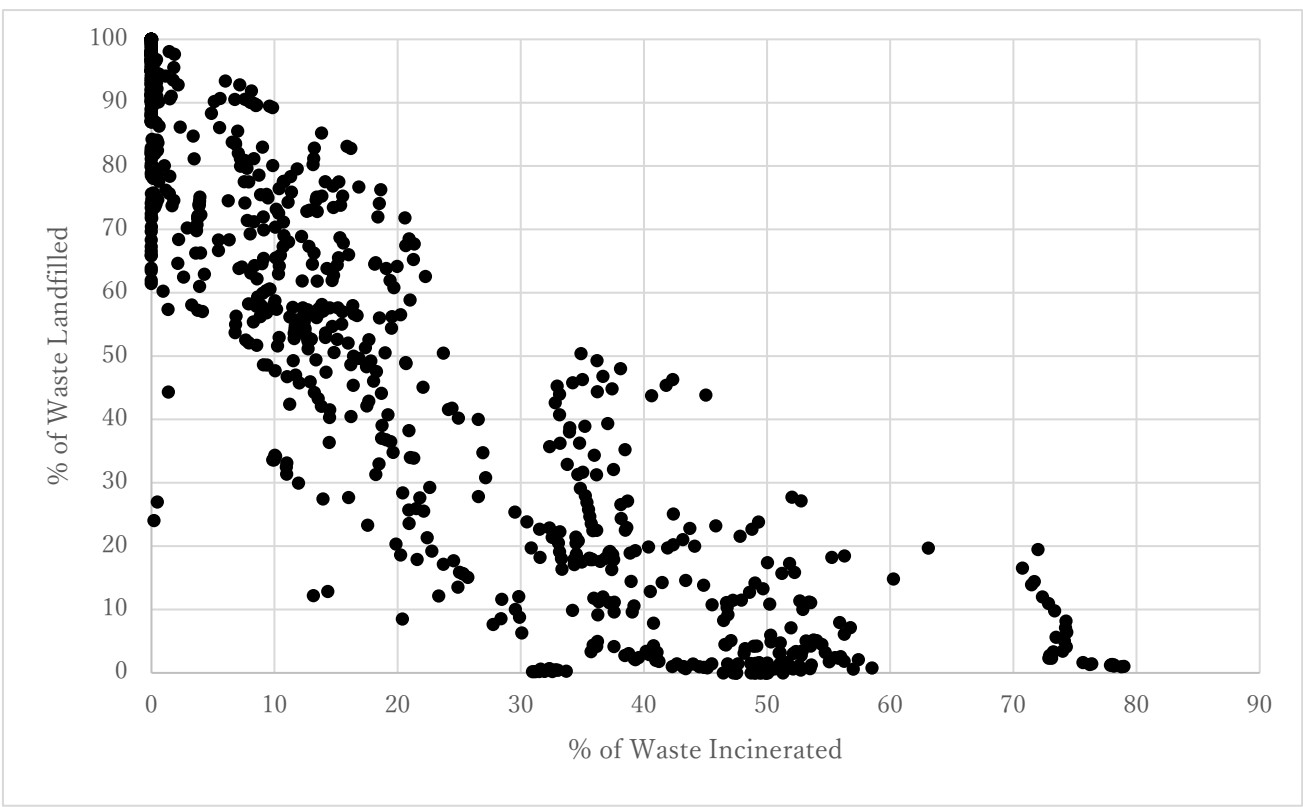

**Figure 1.** Rates of incineration and landfilling.

Figure 2 plots the percentage incinerated versus the percentage recycled for all OECD countries and all years. These data hint at a non-linear correlation between incineration and recycling worthy of further testing below. At low levels of incineration, incineration and recycling appear positively correlated. However, then recycling rates appear to decrease with increases in incineration rates among countries with levels of incineration in excess of about 30%. Recall that the model above demonstrated the rationale for both a positive and negative relationship between incineration capacity and recycling.

The percentage of waste incinerated (on the horizontal axis) and the percentage of waste recycled (on the vertical axis) for each year within each of the three groups of countries is plotted in Figure 3a–c. Figure 3a shows the incineration and recycling rates in Germany, Italy, Korea, and Japan. Each of these countries has experienced a positive correlation over time between incineration and recycling rates. If these correlations are estimated below to be causal, then incineration and recycling would appear to be working in tandem to replace landfilling. Note that among this group of four countries, Japan is unique by experiencing very high rates of incineration and relatively low rates of recycling relative to the other countries in this group.

Figure 3b illustrates the same data but for Norway and Sweden. These two countries experienced a non-linear pattern similar to Figure 2 above. Both countries find incineration and recycling rates to be negatively correlated after incineration rates exceeded roughly 40%. If incineration and recycling were initially working in tandem to replace landfilling, then that working relationship ended once incineration became substantial. Intensive incineration may rely upon recycled materials as fuel to alleviate costly excess capacity.

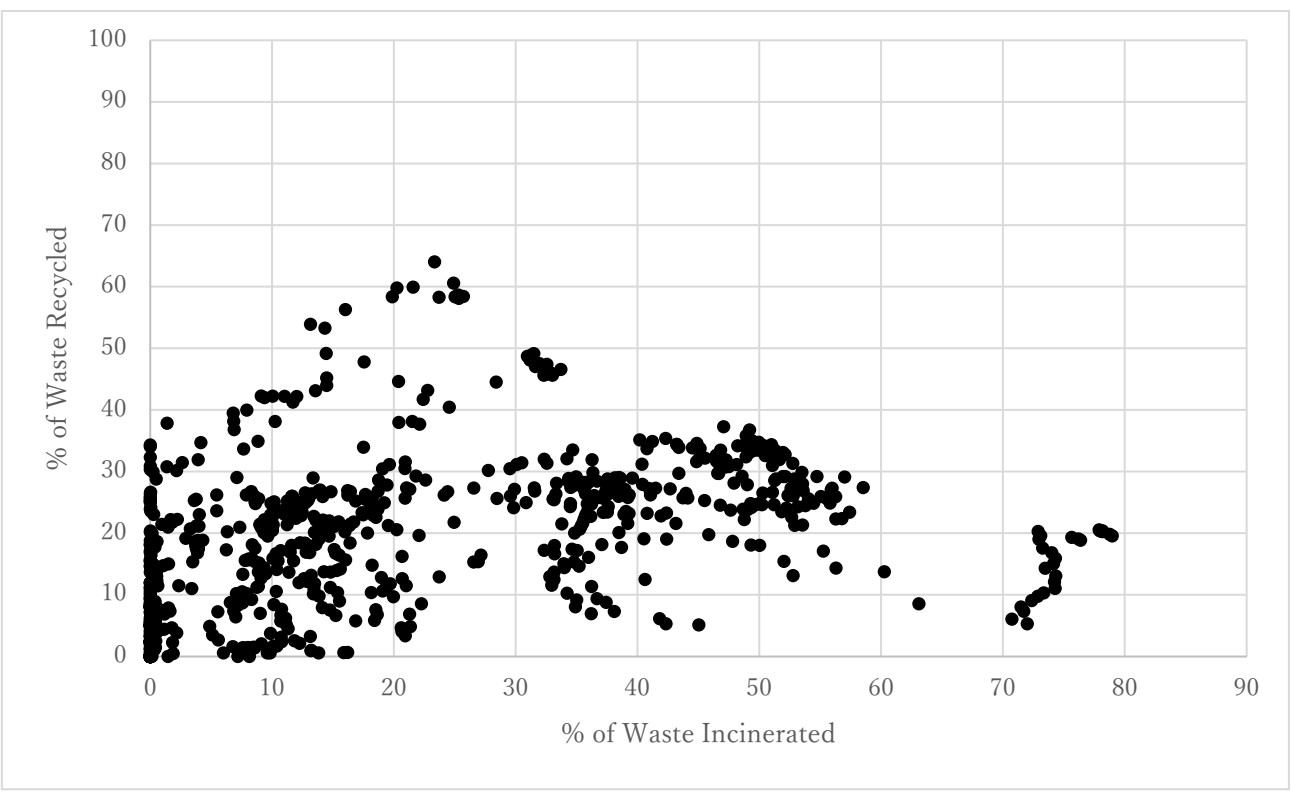

**Figure 2.** Rates of incineration and recycling.

Finally, the situations in Denmark, Luxembourg, and the United States are illustrated in Figure 3c. These three countries experience only a negative relationship between rates of incineration and recycling across all years. Note that both Denmark and Luxembourg experienced large rates of incineration throughout this time span and thus the tradeoff is similar to that experienced at high rates of incineration in Norway and Sweden. The United States is unique in that the inverse correlation between incineration and recycling rates occurs at low rates of incineration. Perhaps in the United States, increases in recycling rates are replacing both incineration and landfilling, perhaps because excess incineration capacity is not rampant.

However, keep in mind that the data patterns emerging in all of these figures are not causal. First, no controls are made for differences in per-capita GDP, population densities, the total amount of waste managed, and other variables that vary across countries but remain constant over time. If any of these variables are correlated with both incineration rates and with landfilling or recycling rates, then the patterns illustrated above may not be causal. More worrisome than omitting control variables is the prospect that these rates of landfilling, recycling, and incineration within countries may be jointly determined. The percentage incinerated relies on the other two forms of waste management and vice-versa. None are exogenous from the others. The next section develops an econometric model to account for these potential problems.

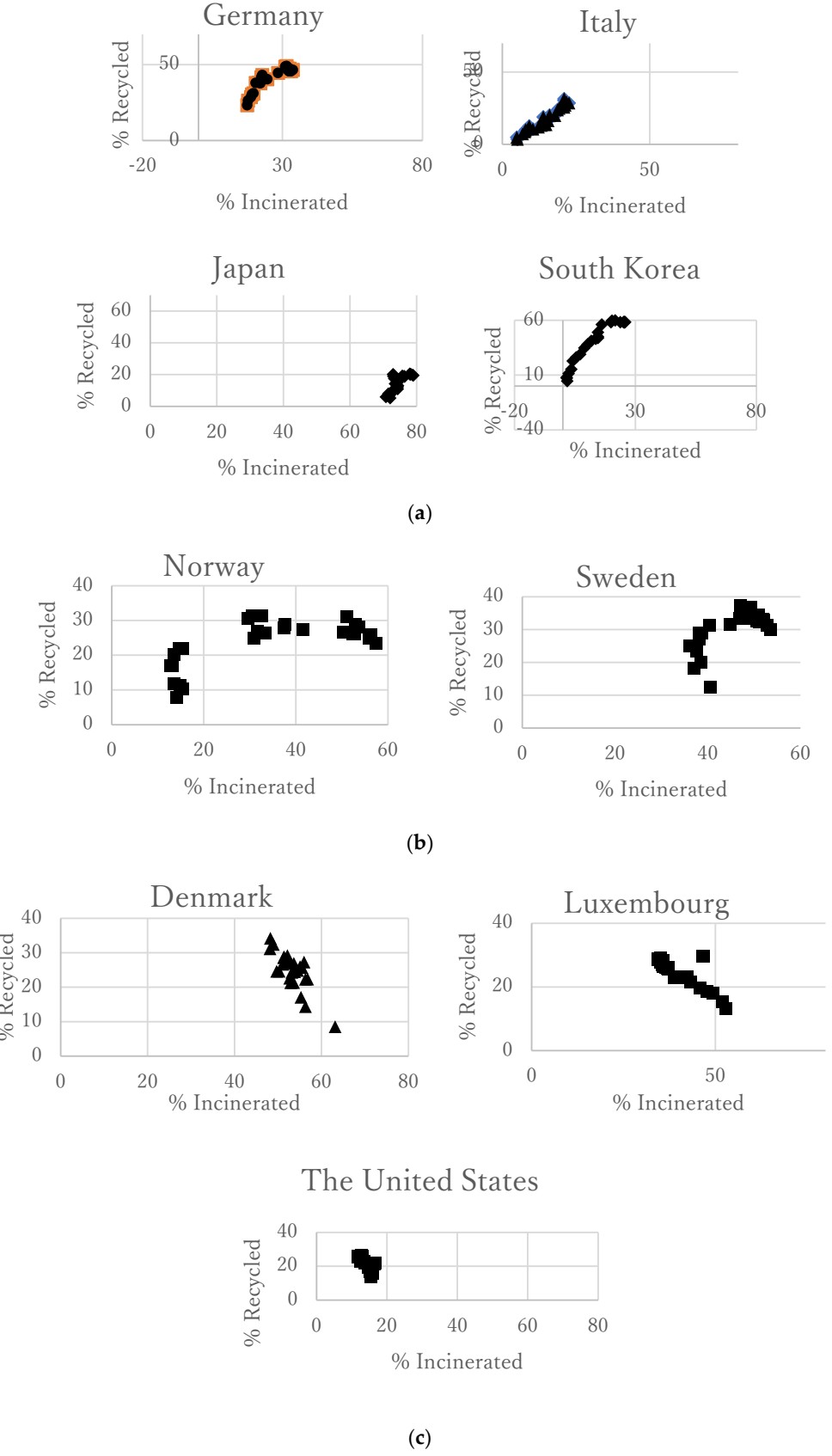

**Figure 3.** (**a**) Countries where incineration and recycling are positively correlated. (**b**) Countries where incineration and recycling are positively and then negatively correlated. (**c**) Countries where incineration and recycling are negatively correlated.

## 5. The Data and the Econometric Model

The econometric model used to estimate the effect of incinerators on each of the three forms of waste processing is below.

$$Q_{I,it} \text{ or } Q_{L,it} \text{ or } Q_{R,it} = \beta_0 + \beta_1 \cdot \ln(\overline{Q}_{it}) + \beta_2 \cdot \ln(pcGDP)_{it} + \beta_3 \cdot \ln(Dens)_{it} + \beta_4 \cdot \ln(Total)_{it} + \beta_4 \ln(pcGDP)_{it} + \beta_5 Year_t + a_i + u_{it}$$

All exogenous variables are logged. A linear time trend is added to control for any unobserved effects on waste that are constant across countries but might change over time. The error term is comprised of unobserved variables that vary across time ($u_{it}$) and do not vary across time ($a_i$). Table 2 provides definitions of each variable in the model.

**Table 2.** Definition of variables.

| Variable | Definition | Mean | Std. Dev. | Minimum | Maximum |
|---|---|---|---|---|---|
| $(Q_I)_{it}$ | The percentage of total waste incinerated in country $i$ in year $t$ | 20.92 | 20.53 | 0 | 78.50 |
| $(Q_R)_{it}$ | The percentage of total waste recycled in country $i$ in year $t$ | 18.35 | 12.82 | 0 | 59.93 |
| $(Q_L)_{it}$ | The percentage of total waste landfilled in country $i$ in year $t$ | 52.39 | 34.33 | 0 | 100 |
| $(\overline{Q})_{it}$ | The number of incinerators in country $i$ in year $t$ | 54.64 | 192.63 | 0 | 1221 |
| $pcGDP_{it}$ | Per-capita income in country $i$ in year $t$ | 29,885.6 | 13,485.89 | 7591.49 | 95,175.73 |
| $Density_{it}$ | Total persons divided by square kilometers in country $i$ in year $t$ | 132.30 | 126.06 | 2.23 | 516.66 |
| $Total_{it}$ | The quantity of all waste collected (in 1000 pounds) in country $i$ in year $t$ | 18,930 | 39,609 | 118 | 238,045 |

The seemingly unrelated regression estimation method is used to estimate these three equations, which will yield estimated coefficients nearly identical to OLS since each equation shares the same list of independent variables. The fixed-effects within the estimator is used on the panel data to eliminate any bias from the unobserved variables that do not vary over time ($a_i$).

Table 2 also provides summary statistics of all variables in the model. The dataset contains values for all 35 countries appearing in Table 1 and is based on a 25-year time span. The mean rates of incineration, recycling, and landfilling are consistent with the values appearing in Table 1. The standard deviation and extreme values suggest once again a wide variety of solid waste management methods across countries and across time within OECD countries. Population density and per-capita GDP also vary substantially across the sample. The lowest observed population density is in Australia in 1990, and the highest is in South Korea in 2014. Per-capita GDP ranges from a 1990 low in Chile to a 2014 high in Luxembourg.

The theoretical model above suggests that a change in incineration capacity could result in a positive or negative effect on recycling. A casual examination of the data above support both possibilities. To test this relationship with the data, we require some measure of incineration capacity, such as the potential tons of combustion capacity per day. Unfortunately, such a precise measure of the total incineration capacity in each country and across each year is not available in the OECD data. Instead, data are only available on the number of solid waste incinerators operating within each country and year. We have no idea if these incinerators are large or small. Nor do we know how much waste can potentially be combusted each day or how much excess incineration capacity is available. Data on the number of incinerators serves as a useful proxy for total incineration capacity only if the average incinerator size within each country and year is uncorrelated with the number of incinerators. If instead countries with a large number of incinerators, they have unusually large or unusually small incinerators, then the estimated coefficient on the number of incinerators will be biased. That coefficient would be telling two stories—the effect of the incinerator itself and the unobserved effect of the unusually large (or small) incinerator. Anecdotally, we have no idea if countries with many incinerators (such as Japan) also have unusually large incinerators or have unusually small incinerators or

have averaged-sized incinerators. The use of panel data and the within estimator helps to mitigate this potential omitted variable bias if countries with many incinerators (and unusually sized incinerators) sustain this relationship over time.

Data on the number of incinerators is available only for the years from 2010 to 2019. Because this variable is our required treatment variable, the panel is effectively shortened to just these years. However, the years 2010 to 2019 experienced a substantial variation in the number of incinerators. For example, the number of operating incinerators increased from 3 to 9 in Finland, from 24 to 42 in the United Kingdom, from 72 to 96 in Germany, from 1 to 7 in Poland, and from 0 to 2 in Ireland. The United States experienced a decrease in the number of incinerators over this time period (from 85 to 73), as did France (from 129 to 121). Many countries did not experience much change in their number of incinerators. Japan and South Korea sustained a very large number of incinerators over this time interval (over 1000 in Japan). Of course, those countries with no incineration, such as Mexico and Turkey and others, had zero incinerators throughout this time span.

The first column of Table 3 reports the estimation fixed-effects results for incineration. Estimates for landfilling and recycling follow in subsequent columns. A one-percent increase in the number of incinerators is estimated to increase the percentage of waste incinerated by 0.047 percent, decrease the percentage of waste landfilled by 0.032 percent, and decrease the percentage of waste recycled by 0.001 percent. Of these three estimates, only the coefficient in the recycling equation is not statistically significant. Thus, an exogenous increase in incineration capacity is estimated to neither increase nor decrease the recycling rate. This result departs from the findings of [2], who estimated that additional incineration capacity reduces recycling rates in Japan. The different results found in this paper and in [2] could be attributable to a number of factors. First, Japan traditionally incinerates roughly 75% of its waste. OECD countries incinerate roughly 25%. Low incineration rates might allow incineration facilities in OECD countries to maintain a consistent flow of waste fuel without having to combust recyclable materials, such as papers and plastics. Second, data in [2] are gathered at the prefecture level from data obtained from individual incinerators. Observed in those data was the quantity of excess incineration capacity, which was found to reduce recycling rates. Unused incineration capacity is not observed in the OECD data—in its place is a proxy variable representing the number of incinerators within a country and year. This difference between treatment variables could explain the differing results.

**Table 3.** Fixed-effects estimation results [1].

|  | $Q_I$ | $Q_L$ | $Q_R$ |
|---|---|---|---|
| $\ln(\overline{Q})$ | 0.047 *** | −0.032 *** | −0.001 |
|  | (0.004) | (0.007) | (0.003) |
| Ln(Per-capita GDP) | 26.45 *** | −49.15 *** | 12.85 *** |
|  | (1.92) | (2.91) | (1.49) |
| Ln(Density) | 2.37 *** | −6.06 *** | 1.24 ** |
|  | (0.74) | (1.12) | (0.57) |
| Ln(Total) | −1.24 ** | 1.32 | 0.46 |
|  | (0.58) | (0.89) | (0.45) |
| Time Trend | 1.14 *** | −2.37 *** | 0.66 *** |
|  | (0.31) | (0.47) | (0.24) |
| Constant | −2381.43 *** | 5008.81 *** | −1368.60 *** |
|  | (624.71) | (949.09) | (484.57) |
|  | N = 271; $R^2$ = 0.57 | N = 271; $R^2$ = 0.59 | N = 271; $R^2$ = 0.25 |

[1] *** denotes statistical significance at the 1% confidence level. ** denotes statistical significance at the 5% confidence level.

A one-percent increase in per-capita income is estimated to increase incineration by 0.26%, decrease landfilling by 0.49%, and decrease recycling by 0.13%. These estimates suggest that increases in per-capita incomes lead to incineration and recycling replacing landfilling for managing waste.

A one percent increase in the population density of a country is estimated to increase incineration by 0.02%, decrease landfilling by 0.06%, and increase recycling by 0.01%. The value of land may increase with population density, and if landfilling operations are more land-intensive than incineration and recycling operations, then the costs of landfilling rises with population density relative to incineration or recycling.

The model also controls for the total amount of solid waste managed in each country. Countries with large total waste quantities may favor those waste management options that enjoy returns to scale. Results here suggest that increases in total waste quantities are estimated to reduce incineration, increase landfilling, and increase recycling, but such changes are only statistically significant in the incineration equation.

Finally, the statistically significant time trend suggests that incineration rates and recycling rates have been rising with time when controlling for the other variables while landfilling rates have been declining with time. The time trend serves as a proxy for unobserved variables that change across all OECD countries with time, such as changes in waste treatment technologies and household preferences for the environment.

**Author Contributions:** Conceptualization, T.C.K. and M.Y.; methodology, T.C.K. and M.Y.; software, T.C.K. and M.Y.; validation, T.C.K. and M.Y.; formal analysis, T.C.K. and M.Y. investigation, T.C.K. and M.Y.; resources, T.C.K. and M.Y.; data curation, T.C.K. and M.Y.; writing—original draft preparation, T.C.K.; writing—review and editing, T.C.K.; visualization, T.C.K. and M.Y.; supervision, T.C.K. and M.Y.; project administration, T.C.K. and M.Y. All authors have read and agreed to the published version of the manuscript.

**Funding:** This research received no external funding.

**Institutional Review Board Statement:** Not applicable.

**Informed Consent Statement:** Not applicable.

**Data Availability Statement:** The supporting data will be provided upon responsible request.

**Conflicts of Interest:** The authors declare no conflict of interest.

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
