# Peer review of "Has Incineration Replaced Recycling? Evidence from OECD Countries"

_sustainability, doi:10.3390/su15043234_

Round 1

Reviewer 1 Report

Referee report on

Has Incineration Replaced Recycling? Evidence from OECD countries

By Thomas Kinnaman/Masashi Yamamoto

Using panel data for 35 OECD countries the relation between incineration, recycling and landfilling is estimated. In an article by the same authors for Japan a negative relationship between incineration and recycling in Japan is found, however, this relation does not appear applicable to OECD countries. As such the data collection with 790 observations, their empirical approach, and their theoretical model is interesting and novel. Nevertheless, the paper contains some elements, which are not convincing:

           First, unbiased coefficients will emerge if unobserved capacity is correlated with the number of incinerators (see line 279), but this is – as mentioned – an unlikely assumption. Nevertheless, it is not clear how for this is corrected? Please put more emphasis on describing the data on the number of incinerators. How are these collected?    

           Second, I am not sure that your theoretical model is so helpful. Costs per municipality are minimized by choosing the quantity incinerated, landfilled, and recycled, but the empirical setting is per country. It is assumed that incineration capacity is exogenously given, but how reasonable is this? First, municipalities must face a decision whether to build a waste-to-energy plant. And why are costs of incineration and recycling not interrelated?  

           Third, In the data paragraph only correlations are given. As mentioned, these are not causal relationships, but in the estimations section only causal relations are estimated by fixed effect estimators (as far as I can check). Especially, I wonder whether Ln(Total) and ?Ì… (see also first point) are exogenously given. Please clarify how you deal with this issue of possible endogeneity or causality.

Small remarks:

p. 1, abstract: “…, an increase in incineration is found to increase rates of incineration …”, this seems trivial to me.

p. 3: “Boesch et al. (2014)” is not mentioned in the reference lists.

p. 5, line 125: Please also discuss the large differences between EU countries (for example Greece with 80% landfilling). And discuss also whether there are relaxation options in an EU directive on landfill.

p. 6, Table 1: As the data for the number of incinerators are only available between 2010 and 2019, I would suggest adding year 2010 in table 1 as well.

p. 12, line 246-247. Please explain: “Thus, the effect of an exogenous increase in incineration capacity (?Ì…) is expected to increase recycling rates”.

p. 15: Please avoid such general statements as for example for plastic waste recycling is more expensive than incineration.  “This result may not be surprising given that modern incineration is often the most expensive of these three waste management practices followed by recycling.”

Author Response

First, unbiased coefficients will emerge if unobserved capacity is correlated with the number of incinerators (see line 279), but this is – as mentioned – an unlikely assumption. Nevertheless, it is not clear how for this is corrected? 

This section has been re-written to discuss the bias and how it was corrected.  Here is an excerpt:

"The model above suggests that a change in incineration capacity could have a positive or negative affect on recycling.  A casual examination of the data above support both possibilities.   To test this relationship with the data, we require some measure of incineration capacity such as the potential tons of combustion capacity per day.  Unfortunately, such a precise measure of the total incineration capacity in each country across each year is not available in the OECD data.  Instead data are only available on the number of solid waste incinerators operating within each country and year.  We have no idea if these incinerators are large or small.  Nor do we know how much waste can potentially be combusted each day.  Data on the number of incinerators serves as a useful proxy for total incineration capacity only if the average incinerator size within each country and year is uncorrelated with the number of incinerators.  If instead countries with a large number of incinerators also have unusually large or unusually small incinerators, then the estimated coefficient on the number of incinerators will be biased.  That coefficient would be telling two stories – the effect of an incinerator itself and the unobserved effect of the unusually large (or small) incinerator.  Anecdotally, we have no idea if countries with many incinerators (such as Japan) also have unusually large incinerators or unusually small incinerators or averaged-sized incinerators.  The use of panel data and the within estimator helps to mitigate this omitted variable bias if countries with many incinerators (and unusually sized incinerators) sustain this relationship over time. "

I am not sure that your theoretical model is so helpful.

In revising the draft, we devoted careful effort to make the model more relevant and helpful to the paper.  The model much better emphasizes the single goal of generating an ambiguous (positive or negative) relationship between incineration capacity and recycling.

Costs per municipality are minimized by choosing the quantity incinerated, landfilled, and recycled, but the empirical setting is per country.

A country’s aggregated waste data are comprised of decisions made at municipal levels in most OECD countries. But if strong federal rules drive decisions at all local jurisdictions, then the model could represent a country’s decision rather than a municipality’s decision.  W added a new Footnote to alert this fact to the reader.

It is assumed that incineration capacity is exogenously given, but how reasonable is this? First, municipalities must face a decision whether to build a waste-to-energy plant.

Assuming exogeneity is often a heavy assumption in econometric modelling. Although we can imagine unobserved variables affecting both the number of incinerators and recycling quantities, we believe these variables are rather constant over time and thus coefficients remain unbiased when using the within fixed effects estimator.  We rely heavily on the idea that incinerators are durable capital assets with fixed capacities in the short run to help explain away simultaneity bias (that recycling quantities are driving the number of incinerators).

And why are costs of incineration and recycling not interrelated?  

The model was developed to serve one purpose – to support the notion that adding incineration capacity may increase or may decrease recycling.  As is the case in all modelling, the goal is to generate a testable hypothesis as simply as possible hoping the simplicity doesn’t ruin the applicability.  Additively separating the components of the cost function was made to keep things simple. Interrelating costs would have added a host of second order interrelated effects that would have weighed heavily on the simplicity of the model without, we expect, changing the main results.  Note also that landfills and recycling facilities and incinerators are often located apart from each other and are sometimes owned/managed by separate companies or arms of government.

I wonder whether Ln(Total) and ?Ì… (see also first point) are exogenously given. Please clarify how you deal with this issue of possible endogeneity or causality. 

We discussed the possible exogeneity of Q-bar above.  Total waste, we believe, is determined by tastes and incomes of households in the country.  Do people generate more waste in response to an added incinerator?  An intriguing thought but not part of this paper.

 “…, an increase in incineration is found to increase rates of incineration …”, this seems trivial to me. and p. 12, line 246-247. Please explain: “Thus, the effect of an exogenous increase in incineration capacity (?Ì…) is expected to increase recycling rates”.

These two comments are related.  The model provides conditions for precisely these two unintuitive points.  The model section has been carefully revised to make this crystal clear to the reader.  Here is a relevant excerpt:

"To summarize, if , then adding incineration capacity increases incineration costs causing incineration to fall and recycling to rise.  But if , then adding excess capacity reduces incineration costs causing incineration quantities rise and recycling quantities fall.  In this latter case, incinerating recyclable materials averts high costs associated with managing unwanted excess incineration capacity.  Thus, the theory provides two possible directions for the causation between incineration and recycling.  What do the data say?"

“Boesch et al. (2014)” is not mentioned in the reference lists.

This reference has been added.

line 125: Please also discuss the large differences between EU countries (for example Greece with 80% landfilling). 

A mention of data on Greece has been added.

And discuss also whether there are relaxation options in an EU directive on landfill.

We added a phrase to remind the reader that such options may exist.  We hesitate to devote significant attention to this policy wrinkle.

Table 1: As the data for the number of incinerators are only available between 2010 and 2019, I would suggest adding year 2010 in table 1 as well.

New columns representing the year 2010 have been added to the Table 1.

Please avoid such general statements as for example for plastic waste recycling is more expensive than incineration.  “This result may not be surprising given that modern incineration is often the most expensive of these three waste management practices followed by recycling.”

The general statement has been removed.

Reviewer 2 Report

In the current research, the author uses panel data to evaluate how incineration influences recycling rates in OECD nations.

The following suggestions were given to the author to include in the work after reviewing the paper.

1.      It is recommended that the author rewrite the abstract section properly, eliminating any grammatical errors.

2.      It is recommended that the author utilize the full form of OECD at least once in the current paper before switching to the short form.

3.      It is urged to the author that they add more recent literature in the article because it is lacking.

4.      Although the author states that "the OECD publishes annual data on the percentage of garbage landfilled, recycled, and burned by each member country each year," the current research only uses data up to 2018. Give a reasonable explanation.

5.      There are numerous grammatical errors in the work; the author has urged to rectify them and, if feasible, rewrite the sentence where necessary.

Author Response

  1. It is recommended that the author rewrite the abstract section properly, eliminating any grammatical errors.

The abstract has been rewritten.

  1. It is recommended that the author utilize the full form of OECD at least once in the current paper before switching to the short form.

The full form of OECD has not been provided in the first line of the paper.

  1. It is urged to the author that they add more recent literature in the article because it is lacking.

We agree that a recent literature review is lacking.  We also recognize that this research question is rather new (only one previous paper).  We connect this research to another literature estimating recycling rates, but recent papers in that literature have not been discovered.

  1. Although the author states that "the OECD publishes annual data on the percentage of garbage landfilled, recycled, and burned by each member country each year," the current research only uses data up to 2018. Give a reasonable explanation.

When the data analysis was conducted for this paper, the 2019 data was not complete – only a small portion of 2019 data was available at that time.  This week, we checked again with OECD to learn that indeed 2019 data are now available but 2020 data is still largely incomplete.  Thus, we could extend effort to adding only 2019 data but feel the main results will not change much (and were only given 10 days by the journal to revise the paper).  Note that the paper does indicate that 2018 was the last year for complete data.

  1. There are numerous grammatical errors in the work; the author has urged to rectify them and, if feasible, rewrite the sentence where necessary.

A fresh editorial eye was given to the entire manuscript with the goal of improving clarity and correcting any grammatical errors.

Reviewer 3 Report

Reviewer’s comments

·         The abstract must be revised, and empirical data are to be included.

·         Manuscript lacks novelty, please clearly state the novelty of your study.

·         Manuscript looks more like a book chapter than a research article, please work on it by consulting other recently published articles in Sustainability.

·         Please check the Guide to authors and use the journal’s template in preparing your manuscript.

·         The service of a language expert must be sought as many sentences were incoherent.

·         Three keywords are inappropriate, at least it can be increased to five.

·         Poorly referenced as reference style deviated completely from journal’s requirement.

·         Although the authors have a good idea in mind, the entire work is poorly presented.

·         Can authors distinguish between the Introduction and Relevant Literatures as both do not really make sense?

·         Manuscript is not numbered based on sections and subsections.

·         Result and Discussions are not adequately presented.

Overall, this manuscript is significant, however, at this stage I do not recommend its publication until the entire manuscript is thoroughly revised and all issues raised are addressed by the authors.

Author Response

The abstract must be revised, and empirical data are to be included.

The abstract has been revised.

Manuscript lacks novelty, please clearly state the novelty of your study.        

The goal of the paper is now stated in both the abstract and the introduction.

Manuscript looks more like a book chapter than a research article, please work on it by consulting other recently published articles in Sustainability. Please check the Guide to authors and use the journal’s template in preparing your manuscript. Poorly referenced as reference style deviated completely from journal’s requirement. Manuscript is not numbered based on sections and subsections.

Other articles published in Sustainability were consulted to help ensure the manuscript fits within the expected style.

Three keywords are inappropriate, at least it can be increased to five.

Two additional keywords have been added

Can authors distinguish between the Introduction and Relevant Literatures as both do not really make sense?

Both of these sections have been revised.

Result and Discussions are not adequately presented.

These sections have been revised

Reviewer 4 Report

The paper addresses an important question about the relationship between incineration/recycling,I found the subject very interesting and think it is a worthwhile topic.  However, I think that you need to address the scientific rigour of some of the work (first half of the paper) as it focusses on the interpretation of results but doesn't pay much attention to the referenced methodologies used to collect data.

Should define who the OECD countries are.

Line 82: You have dropped in the idea of social costs without introducing it properly first.  I would suggest that this is defined more clearly.

The literature findings are reviewed but not the methodolgoies used to deliver those findings.  Some mention of this is important to validate findings as the methods used to obtain the figures will assure the reader of their uesfulness to compare with each other.

Line 124: Use of subnjective terms "do not recycle much at all"; figures should be put to this.

Line 131: The use of a statistical technique, in this case "hint at a non-linear correlation" is introduced but not very scientifically presented. No R or R2 values etc.  The comparison should be purely on the figures (or statistics) not subjective opinion of the relationships.

Line 140: "If this relationship is estimated to be causal" this is not a very scientific way of presenting, comparing, and discussing, this information.

Figures: You need to explain these in more detail including where the information comes from more clearly.  It is not clear to me that these are representing 'rates of incineration and recycling'.

Figure 1: Axis labels should have units of measurment, for the benefit of comparison the y axis should be the same for all.

Figures 3: Y axis label.  Different scales for the x axis makes this information at best, difficult to compare on a like-for-like basis, at worst, a misrepresentation.  Where are the R2 values?

I found the the conclusions interesting, but could have been worded much better for clarity of the message you are giving e.g. "these two circumstances are not incompatible"

Author Response

First half of the paper focusses on the interpretation of results but doesn't pay much attention to the referenced methodologies used to collect data.

The data were all collected by OECD and links to the data tables are provided in the paper.  We expect readers interested in the data source can follow the provided link to learn for themselves about the collection methods rather than devoting a section describing that process in this paper.

Should define who the OECD countries are.

All OECD countries are listed in Table 1.  The OECD acronym is now defined in the first sentence of the paper.

Line 82: You have dropped in the idea of social costs without introducing it properly first.  I would suggest that this is defined more clearly.

This section has now been relocated to hopefully improve the overall organization.  

Line 124: Use of subjective terms "do not recycle much at all"; figures should be put to this.

Precise numbers have been added to this discussion.

Line 131: The use of a statistical technique, in this case "hint at a non-linear correlation" is introduced but not very scientifically presented. No R or R2 values etc.  The comparison should be purely on the figures (or statistics) not subjective opinion of the relationships.

Note that the statistical analysis is not conducted until a later section.  The data are simply introduced in this section to further motivate the research question.  We now make this point clear to the reader. 

Line 140: "If this relationship is estimated to be causal" this is not a very scientific way of presenting, comparing, and discussing, this information.

The appropriate scientific investigation of these data takes place in a later section.  We now make this point clear to the reader. 

Figures: You need to explain these in more detail including where the information comes from more clearly.  It is not clear to me that these are representing 'rates of incineration and recycling'.

The axes of these tables have been revised to clearly define each relevant variable.

Figure 1: Axis labels should have units of measurement, for the benefit of comparison the y axis should be the same for all.  Figures 3: Y axis label.  Different scales for the x axis makes this information at best, difficult to compare on a like-for-like basis, at worst, a misrepresentation.  Where are the R2 values?

Comparisons across countries are probably better made by examining the data in Table 1 rather than these figures.  The purpose of these figures is to understand the general pattern of the relationship between incineration and recycling within each country.  We now explain that in the paper.  Statistical analysis of course takes place in a later section.

I found the conclusions interesting, but could have been worded much better for clarity of the message you are giving e.g. "these two circumstances are not incompatible"

The conclusion has been revised.

Round 2

Reviewer 1 Report

Thank you for revising the paper. I donot have any additional comments. 

Author Response

The entire paper was edited once again to improve clarity and grammar.

Reviewer 2 Report

The revised manuscript version now appears better. Only minor grammar changes are required. 

Author Response

The entire draft was edited by a native English speaker.  Hopefully all grammatical mistakes were detected and corrected.

Reviewer 3 Report

Authors are enjoined to provide responses to reviewers' comments in a professional manner and present the manuscript on the Journal's template.

Author Response

The entire document was edited again by a native English speaker.

Reviewer 4 Report

FIgures 1 goes up to 100% and figure 2 only goes up to 60%.  There is less opportunity for misleading interprestations of the data if they are plotted on the same axis (i.e. both up to 100%).  It may be a small detail, but I think they should have the same label format, i.e. fig 1 has landfill v incinerate, fig 2 has incinerate.  Also, again niether fig 1 or 2 has % , this is a small but important formatting detail that requires little effort to rectify and makes the figures much easier to cross reference for a reader.  Is there an issue with doing this that I am missing?  Figs 3 also are not labelled on at least one y axis.

I agree with the authors point about the patterns, but one could argue that absolute figures are the more appropriate way to compare this between countries, as the Capex and Opex of recycling facilities, or incineration plant, depend so much on the absolute volume/mass of waste. 

I don't believe that it is acceptable to publish paper's that have axis which are not clearly labelled.

Author Response

Figures 1 goes up to 100% and figure 2 only goes up to 60%.  There is less opportunity for misleading interpretations of the data if they are plotted on the same axis (i.e. both up to 100%). 

  • Figures 1 and 2 have been reformatted to contain identical legends 

It may be a small detail, but I think they should have the same label format, i.e. fig 1 has landfill v incinerate, fig 2 has incinerate. 

  • Figures 1 and 2 have been reformatted to contain the appropriately identical axis labels 

Also, again neither fig 1 or 2 has % , this is a small but important formatting detail that requires little effort to rectify and makes the figures much easier to cross reference for a reader.

  • The % sign has been added to all figures.

Figs 3 also are not labelled on at least one y axis.

  • All Figure 3 graphs are now clearly labelled

I agree with the authors point about the patterns, but one could argue that absolute figures are the more appropriate way to compare this between countries

  • The important pattern depicted in each figure is still visible after reformatting to include consistent legends (within each group) and proper labels.  

Round 3

Reviewer 3 Report

1. The manuscript has been significantly improved upon, however the formatting on the Journal's template is poor. This can be improved.

2. Also, the implication and conclusion must be rewritten as the current was just lifted from Reference 1 number.

3. The implication and conclusion may be presented separately while the references at the conclusion may be omitted as it is usually not necessary. 

Author Response

1. The manuscript has been significantly improved upon, however the formatting on the Journal's template is poor. This can be improved.

I followed the template while further revising the manuscript.  Specifically, I removed all Footnotes and made other adjustments.  Note that I usually delay these final editorial changes until after a paper has been accepted by the journal.  Thank you for your patience.

2. Also, the implication and conclusion must be rewritten as the current was just lifted from Reference 1 number.

The conclusion section has been eliminated in the revised draft (the template suggests a conclusion is optional).  The important information from the conclusion - new to this paper - has been moved up into the main body of the paper.  

3. The implication and conclusion may be presented separately while the references at the conclusion may be omitted as it is usually not necessary. 

I believe this issue has been resolved by eliminating the conclusion.